# Pathophysiological Roles of the CX3CL1-CX3CR1 Axis in Renal Disease, Cardiovascular Disease, and Cancer

**DOI:** 10.3390/ijms26115352

**Published:** 2025-06-03

**Authors:** Yuya Iwahashi, Yuko Ishida, Naofumi Mukaida, Toshikazu Kondo

**Affiliations:** 1Department of Forensic Medicine, Wakayama Medical University, 811-1 Kimiidera, Wakayama 641-8509, Japan; 0911yuya@wakayama-med.ac.jp (Y.I.); mnao.otcyt@gmail.com (N.M.); 2Department of Urology, Wakayama Medical University, Wakayama 641-8509, Japan

**Keywords:** CX3CL1, CX3CR1, inflammatory diseases, renal diseases, cardiovascular diseases, cancers

## Abstract

CX3CL1 and its unique receptor, CX3CR1, are leukocyte migration factors and are involved in the pathogenesis and progression of many inflammatory diseases and malignancies. The CX3CL1-CX3CR1 axis induces a variety of responses, including cell proliferation, migration, invasion, and apoptosis resistance. CX3CL1 is a transmembrane protein, and proteolysis generates a soluble form. The membrane and soluble forms of CX3CL1 exhibit different functions, but both bind to the chemokine receptor CX3CR1. The CX3CL1-CX3CR1 axis is a chemokine system that has attracted attention not only as a therapeutic target but also as a potentially useful diagnostic and prognostic marker for disease. Many studies have reported that the CX3CL1-CX3CR1 axis is involved in disease progression, but more recently there are scattered reports suggesting that it is involved in disease suppression. In this article, we summarize the latest findings on the pathophysiological role of the CX3CL1-CX3CR1 axis, with a particular focus on renal disease, cardiovascular disease, and cancer.

## 1. Introduction

CX3CR1 is a seven-transmembrane G protein-coupled receptor that is ubiquitously expressed on mononuclear cells and circulating lymphocyte leukocytes in most tissues [1]. Fractalkine, also known as CX3CL1, discovered in 1997, is the only ligand for CX3CR1 and the chemokine that not only involves chemoattractant functions but also assists CX3CR1+ cell adhesion [2,3]. CX3CL1 exists in two forms: a transmembrane protein that interacts with its own CX3CR1 receptor and acts as an adhesion molecule, and a soluble protein that is generated by proteolytic processes and acts as a signaling molecule [4,5]. Fractalkine molecule is composed of a total of 373 amino acids. Fractalkine is a unique transmembrane chemokine molecule because it is divided into four segments and synthesized across the membrane [6]. The first extracellular component of fractalkine consists of two parts: an N-terminal domain consisting of 76 amino acids and a mucin-like stalk containing 241 amino acids. The extracellular segment of the fractalkine molecule is attached to a transmembrane alpha-helix composed of 19 amino acids in the membrane space and finally completes its model with an intracytoplasmic tail composed of 37 amino acids [6]. Extracellular segments of fractalkine molecules are cleaved by tumor necrosis factor alpha-converting enzyme (TACE or ADAM17) [7] and disintegrin-like metalloproteinase 10 (ADAM10). This produces a soluble component of fractalkine (sFKN), which allows chemoattraction of immune cells such as monocytes, NK cells, and T cells [8]. ADAM10 organizes cleavage during homeostasis [9] and ADAM17 has been found to be used to facilitate the process during the inflammatory phase, which is stimulated by drugs such as lipopolysaccharide (LPA) and interleukin-1β (IL-1β) [7]. Furthermore, ADAM is regulated by TIMPs, which inhibit and regulate the release of plasma membrane proteins [10].

Regarding leukocyte migration, it usually begins with the trapping and rolling of leukocytes on the endothelial surface with E-, L-, and P-selectins [11]. Once these immune cells adhere to the endothelial surface, integrins are activated, ensuring leukocyte adhesion to the endothelium. This is further helped by its interaction with intracellular adhesion molecule (ICAM)-1, a ligand expressed in the endothelium [12]. The captured leukocytes then exit the vessel wall and migrate to the injured tissue. In the presence of fractalkine, fractalkine and CX3CR1 bind with high affinity, eliminating the need for integrin and selectin molecules, which are no longer required for leukocyte adhesion to the vessel wall [13,14]. Fractalkine has a high-affinity binding capacity for CX3CR1 receptors, allowing faster binding, stronger adhesion, and circumvention of the classical migration system that moves leukocytes through the vessel wall [15]. It can skip the rolling phase of leukocyte migration [16], replaces the need for VCAM-1 for integrin binding, and is also thought to speed up the integrin-mediated leukocyte migration process [17,18]. After CX3CL1 binds to CX3CR1, the CX3CL1/CX3CR1 axis can initiate a cascade through several signaling pathways, including ROS/MAPKS, Raf/MEK1/2-ERK1/2-Akt/PI3K, and NF-kB [19]. The CX3CL1-CX3CR1 axis has been shown to be involved in a variety of diseases (Figure 1).

Over the past two decades, monoclonal antibodies targeting proinflammatory chemokines or their receptors have made a great progress in the treatment of many diseases, including malignant tumors and autoimmune diseases. The potential of targeting the CX3CL1-CX3CR1 axis in inflammatory or fibrogenic diseases has been suggested [20,21]. The CX3CL1-CX3CR1 axis is an attractive therapeutic target because it is composed of a unique ligand-receptor pair that does not interact with other partners, limiting the possibility of off-target effects [22]. Meanwhile, the CX3CL1-CX3CR1 axis is differentially regulated in various diseases. In many cardiovascular diseases, elevated plasma CX3CL1/fractalkine concentrations increased platelet activation and the formation of platelet–monocyte complexes, eventually resulting in the promotion of atherosclerosis. However, little progress has been made in this field so far [20,23], and clinical trials targeting CX3CL1-CX3CR1 in cardiovascular disease have not progressed so far. A further understanding of the early stages of atherosclerosis development is needed to design specific drugs that target CX3CL1-CX3CR1 and prevent the progression of atherosclerosis and ideally the development of atherosclerosis itself. In addition, tumors have a strong systemic aspect, as metastases occur in the majority of patients. The efficacy of existing treatments is far from satisfactory [24,25]. To date, CX3CL1 has been associated with numerous malignancies. Therefore, it is important to understand the exact mechanism of action in various disease situations (tumor malignancies, autoimmune diseases, neurodegenerative diseases, etc.). This review summarizes the pathophysiological roles of the CX3CL1-CX3CR1 axis, with a particular focus on renal disease, cardiovascular disease, and cancer.

## 2. Renal Diseases

### 2.1. Renal Injury and Inflammation

Renal inflammation is a complex biological process that plays a critical role in promoting tissue damage and fibrosis in various kidney diseases. Upon injury, an intricate interaction between renal parenchymal cells and resident immune components sets off early inflammatory signaling, promoting the infiltration of circulating leukocytes [26]. These leukocytes, once activated, influence the progression and resolution of injury through a dynamic balance between proinflammatory and anti-inflammatory mediators. An imbalance in this regulatory network can result in persistent inflammation, ultimately leading to renal fibrosis and functional deterioration.

Among the chemokines implicated in both acute and chronic renal inflammation, CX3CL1 plays a significant role [27]. Investigating its expression patterns and downstream signaling can provide deeper insights into the molecular basis of renal pathophysiology. In this context, the molecular mechanisms governing TNF-α–induced CX3CL1 expression in mesangial cells were first explored by Chen et al. [28]. They demonstrated that stimulation of mesangial cells with TNF-α leads to an upsurge in CX3CL1 expression at both transcript and protein levels. The resulting protein undergoes cleavage by matrix metalloproteinases (MMPs), allowing it to facilitate transmigration of monocytic cells. Further analysis using inhibitors targeting protein kinases and the NF-κB pathway revealed that blocking protein kinase C (PKC), ERK1/2 (p42/44 MAPK), NF-κB, or activation protein-1 (AP-1) led to diminished CX3CL1 expression following TNF-α stimulation. These findings collectively highlight that TNF-α promotes CX3CL1 expression through a convergence of intracellular pathways involving PKC, ERK1/2 MAPK, NF-κB, and AP-1 [28].

### 2.2. Glomerulonephritis

Immunoglobulin A nephropathy (IgAN) is the most common primary glomerulonephritis worldwide. Approximately 20–40% of patients will develop end-stage renal disease (ESRD) by 20 years after biopsy [29]. The clinical prognosis of patients with IgAN is often highly variable, as IgAN presents with a very diverse range of clinical manifestations, from asymptomatic urinary abnormalities to rapidly progressive glomerulonephritis. Therefore, the identification of prognostic markers may lead to an improved prognosis for IgAN patients. A previous study investigated serum CX3CL1 levels and renal prognosis in Chinese patients with IgA nephropathy [30]. In patients with IgAN, CX3CL1 was predominantly negatively correlated with creatinine level. Mesangial hypercellularity was also significantly correlated with plasma CX3CL1 level. In renal biopsies of IgAN patients, the number of CD20^+^ B cells and CD68^+^ macrophages correlated significantly with plasma CX3CL1 level, but not with CD4^+^ or CD8^+^ T cells. These findings suggest that CX3CL1 is involved in renal injury by promoting inflammatory cell infiltration. In vitro experiments also showed that when mesangial cells were stimulated with CX3CL1, CX3CL1 not only promoted macrophage cell migration but also directly induced extracellular matrix synthesis in mesangial cells. In another report, stimulation of peripheral blood mononuclear cells from patients with IgA nephropathy with lipopolysaccharides increased the protein and gene expression of CX3CR1 compared to healthy controls, and cells from patients with active disease were more sensitive to CX3CL1 [31].

In a prospective study measuring urinary markers as a prognostic factor for other glomerulonephritis conditions, such as crescentic glomerulonephritis, a multivariate analysis reported that the urinary CX3CL1/urinary creatinine ratio (spot urine protein to creatinine ratio) tended to be associated with good prognosis [32]. In 2022, it was reported that CD16^+^ monocytes were significantly decreased in the blood but increased in glomeruli of patients with myeloperoxidase-antineutrophil cytoplasmic antibody (ANCA)-associated vasculitis (MPO-AAV) compared with healthy controls. Serum CX3CL1 level, but not CCL2 level, was also significantly elevated in MPO-AAV patients. Thus, it is likely that MPO-ANCA increases CX3CL1 expression in human glomerular endothelial cells (HGECs) and promotes the recruitment of CD16^+^ monocytes to the kidney. Enhanced extravascular migration of CD16^+^ monocytes to the kidney via the CX3CL1-CX3CR1 axis may be involved in renal injury [33]. In 2002, Cockwell et al. examined CX3CL1 expression in renal biopsies of patients with acute glomerulonephritis due to AAV. At the gene expression level, CX3CL1 mRNA was increased in the glomerular and tubular interstitium of patients with acute AAV. Immunohistological examination showed that CX3CL1 expression was increased where T cells or macrophages infiltrated [34].

Investigations on glomerulonephritis have also been carried out in animals; MRL/lpr mice spontaneously develop glomerulonephritis with increased circulating immune complexes, autoantibody production, and cytokine abnormalities [35]. The glomerular lesions seen in MRL/lpr mice are diffuse, cell proliferative and/or wire loop-like lesions and are a frequently used animal model, as they resemble the various histological patterns observed in human lupus nephritis [36,37,38]. CX3CL1 expression and CD16^+^ monocyte accumulation in glomeruli were associated with histopathological features of glomerular lesions in MRL/lpr mice [39]. In addition, lupus nephritis showed functional and histological improvement after treatment with anti-CX3CL1, indicating a therapeutic effect of anti-CX3CL1 [39].

Considering all this, it seems that suppressing CX3CR1 can ameliorate glomerulonephritis, but surprisingly, CX3CR1-deficient mice, both male and female, had exacerbated lupus nephritis [40], suggesting that it also plays an important role in suppressing inflammation associated with systemic lupus erythematosus (SLE). The results of this study suggest that CX3CR1 is a key regulator of lupus nephritis. Lack of Cx3cr1 expression was associated with significantly altered intestinal microbiota composition, which was linked to an impaired intestinal barrier [41]. However, none of the previous studies had completely knocked out the gene; instead, antagonism or antibody neutralization was used, which may not completely remove the protein. This suggests a potential dose-dependent effect of CX3CR1 in SLE, with exacerbation of disease at a high dose, whereas a low dose may actually be protective [40]. This finding reflects the complexity of the various pathways that the CX3CL1/CX3CR1 axis regulates in leukocyte migration and may be very important in the therapeutic target for the CX3CL1/CX3CR1 axis.

### 2.3. Chronic Kidney Disease

Plasma CX3CL1 was elevated in chronic kidney disease (CKD) patients and negatively correlated with estimated glomerular filtration rate (eGFR), suggesting a relationship between CKD and the CX3CL1/CX3CR1 axis [42,43]. A common pathogenetic mechanism in CKD is interstitial fibrosis and inflammatory cell infiltration renal fibrosis is characterized by excessive extracellular matrix deposition and myofibroblast accumulation [44]. Previous study reported that the origin of myofibroblasts in the kidney fibrosis was 50% arising from local resident fibroblasts and 35% from bone marrow [45]. Another report showed that 60% of α-SMA-positive myofibroblasts derived from bone-marrow derived cells via the process of macrophage-myofibroblast transition [46]. The origin of myofibroblasts remains controversial.

Comparing with CX3CR1 expression within the renal parenchyma in patients with a variable degree of interstitial scarring of more than 40% and normal kidneys found that CX3CR1 was rarely expressed in normal kidneys, while CX3CR1^+^ cells were identified in fibrotic kidneys within the tubulointerstitium of the medulla and cortex [47]. Serial sectioning showed that CX3CR1 positive cells existed within only CD68^+^ monocytes/macrophages or CD3^+^ lymphocytes. Double immunofluorescence staining showed that tubular epithelial cells, myofibroblasts, and dendritic cells (DCs) also express CX3CR1 in human fibrotic kidneys [47]. Furthermore, human cortical fibroblast cell lines also express CX3CR1 and migrate towards the CX3CL1 in dose-dependent manner, suggesting a functional role for CX3CL1/CX3CR1 in promoting renal fibrosis through fibroblast recruitment [47]. Kassianos et al. have reported that DCs are increased in human CKD by flow cytometric analysis. In particular, the CD141 (hi) and CD1c^+^ myeloid DC subsets mainly produced TGFβ [48]. Furthermore, CD1c^+^ DCs express CX3CR1 and are the main source of TGF-β production in CKD. In addition, CX3CL1 derived from proximal tubular epithelial cells recruited CD1c^+^ DCs into the renal tubulointerstitium and was involved in the development of fibrosis and the progression of CKD [49]. On the other hand, other reports have focused on monocyte subsets; Cormican et al. investigated the circulating intermediate monocytes (IMs) repertoire abnormalities in CKD patients [42]. The IMs of major histocompatibility complex class II (MHC class II) high was significantly increased in CKD patients compared to healthy patients. When these MHC class II high IMs were stimulated with LPS, they produced more proinflammatory cytokines (TNF, IL-1β) than other subsets. Migration toward CX3CL1 and adhesion to primary endothelial cell layers were increased in MHC class II high IMs with CKD compared with healthy patients. It is known that monocyte adhesion to endothelial cells is partially dependent on CX3CR1/CX3CL1 interactions [50]. Thus, CX3CR1 is expressed on dendritic cells and macrophages of specific subset and is involved in renal fibrosis.

Furuichi et al. employed ischemia–reperfusion injury (IRI) in mice, a model of the transition from acute kidney injury to chronic renal failure, to assess the importance of macrophage infiltration in the CX3CL1/CX3CR1 axis [51]. The location of CX3CL1 expression within the kidneys changed, being limited to endothelial cells in sham-operated animals while also detected on tubular epithelial cells and in association with infiltrating cells following ischemia–reperfusion injury. *Cx3xr1^−/−^* mice showed significantly reduced collagen deposition and macrophage infiltration in kidney. Similar results were obtained by administering CX3CR1-neutralizing antibody after IRI, indicating that the CX3CL1/CX3CR1 axis is a potential therapeutic target for fibrosis following renal IRI [51].

The CX3CL1/CX3CR1 axis is not only involved in cell migration but also in cell apoptosis [52]. Lyszkiewcz et al. showed that expression of CX3CR1 promoted the generation of DCs and monocytes/macrophages under steady-state conditions. In a competitive adoptive transfer model, CX3CR1-deficient dendritic cells, monocytes, macrophages, and granulocytes are disadvantages for survival. These effects are not observed in animals exposed to sub-lethal irradiation, suggesting that the survival disadvantage conferred by CX3CR1 deficiency is overcome in inflammatory conditions [52]. On the other hand, renal fibrosis was suppressed in *Cx3xr1^−/−^* mice compared to WT mice using a unilateral ureteral ligation mice model that strongly induces renal fibrosis. This is not because the CX3CL1/CX3CR1 axis is involved in macrophage migration, but because it controls the survival of Ly6C^−^CX3CR1^high^ macrophages, which is important for fibrosis [53]. Engel et al. also showed that a lack of CX3CR1 increases apoptosis of renal macrophages in a mouse unilateral ureter ligation (UUO) model [54]. These observations indicate that CX3CR1 may also affect monocyte/macrophages survival in inflammatory conditions. Notably, renal fibrosis was worse in *Cx3xr1^−/−^* mice [54]. Therefore, it is debatable whether the CX3CL1/CX3CR1 axis contributes to renal fibrosis. Whether this axis is also an important, targetable mechanism of acute kidney injury (AKI) progression to CKD in humans remains less understood and requires more detailed investigation.

### 2.4. Renal Allograft Rejection

In renal transplantation, renal allograft rejection is one of the major complications affecting its prognosis. Early detection of renal allograft rejection and initiation of treatment may lead to good renal prognosis, and the identification of markers that can detect renal allograft rejection at an early stage is of great clinical importance. There are several reports that CX3CL1 and CX3CR1 levels in the serum and urine can be useful predictors of acute renal allograft rejection after the transplantation [55,56,57]. It is known that CX3CL1 induces the migration and adhesion of leucocytes [27]. Mononuclear cell infiltration into renal tubular sites and associated tubular epithelial cell damage are key events in acute renal inflammation after renal transplantation [58]. Hoffmann et al. revealed the expression, distribution, and cellular localization of CX3CR1 in human renal transplant biopsies. The number of CX3CR1^+^ interstitial cells was significantly higher in patients with acute tubulointerstitial and acute vascular rejection compared to normal renal allograft biopsies. Furthermore, CX3CR1^+^ cells were mainly CD68 positive monocytes/macrophages and CD209/DC-SIGN^+^ dendritic cells, suggesting that these cells may play an important role via the CX3CL1/CX3CR1 axis in acute renal transplant rejection [59].

A prospective cohort study examined the clinical relevance and optimal timing of the measurement of urinary cytokines/chemokines as prognostic markers of the intrarenal immunologic micromilieu after living donor renal transplantation. In addition to CCL2/MCP-1, CX3CL1, TNF-α, RANTES, and IL-6 were significantly higher in the early acute rejection group within 3 months after surgery [55]. Krupickova et al., when monitoring the blood levels of 12 chemokines (CXCL1, CXCL5, CXCL6, CXCL8, CXXCL9, CXCL10, CXXCL11, CXCL16, CCL2, CCL5, CCL21, and CX3CL1) in renal allograft transplantation, found that pre-transplant blood levels of CX3CL1 and CXCL10 were particularly high in patients with acute rejection, suggesting a higher inflammatory state compared to uncomplicated patients [60]. A retrospective study found that the combination of CX3CL1 on day 0, IP-10 on the 7th day, and IFN-γ on the 7th day had the highest area under receiver operator characteristics (ROC) curve (AUC) in predicting acute rejection in renal transplant patients, with a sensitivity of 86.8% and specificity of 89.8% [56]. In patients with elevated serum levels of CX3CL1 and CX3CR1, three patients had a pathological diagnosis of early AR, without signs of elevated serum creatinine or proteinuria. Serum CX3CL1 and CX3CR1 concentrations were suggested to be more sensitive than elevated serum creatinine in the diagnosis of early acute rejection (AR) [57]. Furthermore, it has been reported that administration of anti-CX3CR1 antibodies improved survival after heart transplantation in mice [61]. It is therefore possible that the CX3CL1/CX3CR1 axis may be an effective therapeutic target in renal transplant rejection.

### 2.5. Renal Infection Diseases

Chousterman et al. showed that inflammatory monocytes have a protective effect against renal sepsis via a CX3CR1-dependent adhesion mechanism [62]. During sepsis caused by cecal ligation and puncture (CLP)-induced polymicrobial sepsis animal model, inflammatory monocytes migrated from the bone marrow, approached renal cortical endothelial cells and stimulated monocytes within few hours by increasing CX3CR1-related adhesion. Deficiency of CX3CR1 increased renal injury and reduced mortality in mice, which was associated with monocyte migration. The protective function of CX3CR1 was also found to be associated with reduced adhesion of inflammatory monocytes and IL-1ra secretion by Ly6C^high^ monocytes. On the other hand, CX3CR1 is also involved in the pathogenesis of sepsis in human, and a study on polymorphisms in the CX3CR1 gene showed that the I249 CX3CR1 allele correlated with more monocyte adhesion and less renal damage [62]. We also observed an increased mortality in *Cx3cr1^−/−^* mice compared to wild-type (WT) mice with CLP, although the degree of intraperitoneal leukemic infiltration was comparable to that in WT mice [63]. CX3CL1 enhanced gene expression of IL-1β, TNF-α, IFN-γ and IL-12 by WT macrophages, as well as activation of NF-κB. Thus, CX3CL1/CX3CR1 axis was crucial for optimal host defense against bacterial infection by activating bacterial killing function of phagocytes and enhancing iNOS-mediated NO production and bactericidal proinflammatory cytokine production, mainly through the NF-κB signaling pathway [63]. In studies using serum from patients admitted to intensive care units (ICU) with sepsis, serum CX3CL1 was significantly higher in septic patients than in healthy patients [64,65]. Furthermore, serum concentrations of CX3CL1 were significantly correlated with the number of sepsis survivors. Blood leukemia cell counts and inflammatory cytokines (IL-6, IL-1β, IL-17A, IFN-γ, and TNF-α) were positively correlated with blood CX3CL1, while they were negatively correlated with the anti-inflammatory cytokine IL-10. In a study using a mouse model of CLP-induced sepsis, CX3CL1 treatment increased mortality and inflammatory cytokines. Mortality rate was lower in CX3CL1^−/−^ mice and, furthermore, there was less liver, kidney, and lung damage. CX3CL1 reduced bacterial clearance by decreasing the phagocytic capacity of macrophages and neutrophils and the intracellular clearance of *Escherichia coli* (*E. coli*). This suggested that CX3CL1 may exacerbate sepsis by increasing inflammation and decreasing bacterial clearance [65]. The involvement of CX3CL1/CX3CR1 axis in sepsis, therefore, has been reported conflictingly and requires further investigation.

Resident macrophages accumulated in the kidneys, the main target organ of infection, and came into direct contact with the fungus, mainly within the first few hours after infection in a mouse model of systemic Candida albicans infection. Both macrophage accumulation and contact with the Candida was significantly reduced and macrophage survival was reduced in *Cx3cr1^−/−^* mice [66]. In humans, the dysfunctional CX3CR1 allele *CX3CR1-M280* was associated with increased risk of systemic candidiasis. This demonstrated that CX3CR1 is important for innate host defense against the most common fungal pathogens, acting by promoting macrophage survival and accumulation in tissues, and is associated with early and efficient contact between tissue-resident macrophages and fungi, and control of Candida growth and immune evasion [66]. The CX3CL1/CX3CR1 axis has also been shown not to be essential for mucosal infection, indicating that different factors are involved in the control of mucosal candidiasis and systemic candidiasis [67].

### 2.6. Diabetic Nephropathy

Diabetic kidney disease (DKD) is a major cause of end-stage renal disease (ESRD), affecting approximately 20–50% of all diabetic patients [68]. Patients with ESRD require dialysis, which is directly linked to a reduced quality of life and an increased burden on the overall healthcare economy. Early treatment of DKD is important to prevent progression to ESRD. In 1991, Hasegawa et al. first proposed the possibility that inflammatory factors such as TNF and IL-1 are involved in DKD [69]. Subsequently, several reports have supported the association between inflammatory cell infiltration and the progression of DKD [70,71]. These studies have suggested that the CX3CL1/CX3CR1 axis, which is implicated in inflammatory factors, may be associated with DKD, although there are currently only a few reports on this subject. Recently, Chen et al. extracted gene expression data from the Gene Expression Omnibus (GEO) database and validated diabetic tubulopathy genes using Weighted correlation network analysis (WGCNA) and machine learning. They finally identified three hub genes (*FSTL1*, *CX3CR1*, and *AfiR2*), which were negatively correlated with GFR. As a validation study, they tested these genes in mice in streptozotocin (STZ)-induced and *ob/ob* DKD mouse models and found that the three hub genes were increased [72].

Wu et al. similarly examined important genes associated with DKD using GEO database by WGCNA. Several genes were eventually identified as candidate biomarkers, one of which was CX3CR1 [73]. CX3CR1 was shown to be potentially useful as a diagnostic marker for diabetic nephropathy and a predictor of disease progression. The prevalence of diabetes mellitus (DM) was significantly higher in the high CX3CL1 group in an adjusted model in a study of prospectively collected serum samples from CKD patients [74]. Moreover, a previous study has demonstrated that CX3CL1 is induced in inflamed human adipose and that human adipocytes support monocyte adhesion via CX3CL1 [75]. This suggests that CX3CL1 may regulate glucose homeostasis and insulin resistance. In a report using STZ-induced DKD rats, CX3CL1 and CX3CR1 mRNA was significantly increased in the early stages of diabetic kidney compared to controls. Furthermore, this increased expression was suppressed by angiotensin-converting enzyme inhibitor and aminoguanidine administration [76]. Taken together, the evidence indicates that the CX3CL1/CX3CR1 axis may be a useful marker and predictor of treatment response in DKD (Table 1).

## 3. Cardiovascular Disease

### 3.1. Atherosclerosis

The formation of atherosclerosis is associated with vascular endothelial cell injury. The response-to-injury hypothesis proposed by Ross and Glomset suggests that the development of atherosclerotic lesions is the outcome of injury to the arterial endothelium that initiates the interaction of cellular populations of peripheral blood with cell components of the arterial wall [77]. In the early stages, inflammatory cells such as monocytes and T lymphocytes infiltrate when arterial endothelium are damaged. Monocytes that infiltrate the endothelium transform into macrophages, which phagocytose low density lipoprotein (LDL) accumulated in the vascular endothelium and become foam cells. Activation of macrophages and T lymphocytes in plaques causes them to secrete cytokines, which induce plaque rupture. CX3CL1/CX3CR1 promotes monocyte recruitment and inflammatory cytokine release in atherosclerosis and contributes to the development of atherosclerosis at various stages, including vascular smooth muscle cell migration and angiogenesis. Wong et al. studied the coronary arteries of young patients who had died as the result of acute trauma. Plaques expressing CX3CL1 were analyzed by immunohistochemistry. They found that CX3CL1 is expressed at all stages of atherosclerotic lesion formation, and that the number of CX3CL1-expressing cells positively correlates with the number of CX3CR1-positive cells in human carotid artery plaques [78]. CX3CL1 was expressed at both early and advanced stages of atherosclerotic formation, using human atherosclerotic plaque and blood samples from patients with carotid artery disease undergoing endarterectomy. CX3CL1 was expressed in smooth muscle cells, endothelium, neovessels, and macrophages. The number of CX3CL1-expressing cells was positively correlated with the number of CX3CR1-positive cells in carotid plaques. In addition, soluble CX3CL1 levels in circulating blood were significantly elevated in the presence of severe stenosis of the carotid artery [79]. Saederup et al. showed that *Ccr2^−/−^* and *Cx3cl1*^−/−^ mice had significantly reduced macrophage accumulation and atherosclerotic lesion size compared to mice lacking each gene alone. In addition, *Cx3cl1*^−/−^ mice did not have reduced numbers of circulating monocytes, regardless of CCR2 deficiency. CCR2 and CX3CL1 are independently involved in the formation of atherosclerotic lesions. They further state that multiple chemokines or chemokine receptors should be targeted in the treatment of atherosclerosis [80]. In another study, combined inhibition of CCL2, CX3CR1, and CCR5 in *Apoe*-deficient mice resulted in the abrogation of bone marrow monocytosis and reduction in circulating monocytes, despite persistent hypercholesterolemia. The size of the atherosclerotic lesion correlated with the number of circulating monocytes of the CD11b^+^Ly6G^-^Ly6C^lo^ subset. Combined inhibition of chemokines resulted in 90% reduction in atherosclerosis, supporting the usefulness of combined inhibition of chemokines [81]. In a meta-analysis of genetic polymorphisms of CX3CR1, 280M allele carriers of the CX3CR1 T280M polymorphism had a reduced risk of atherosclerosis and coronary artery disease in the heterozygous state but an increased risk of ischemic cerebrovascular disease in the homozygous state. The 249I allele carriers of the CX3CR1 V249I polymorphism had a reduced risk of atherosclerosis and coronary artery disease in the heterozygous state. The V249I-T280M combined genotypes also reduced the risk of atherosclerosis [82].

We have reported explorations of the involvement of the chemokine system in cardiovascular disease in mouse models of aortic aneurysm, aortic dissection, and deep vein thrombosis [83,84,85,86]. In the formation of aortic aneurysms and aortic dissection, as in atherosclerosis, chemokine system is closely involved. Gene expression of CX3CL1 and CX3CR1 levels are elevated in lesions of aortic aneurysm and aortic dissection, and CX3CL1/CX3CR1 may be involved in these diseases as well, requiring further investigation. Deep vein thrombosis was examined in a model of the ligation of the inferior vena cava. *Cx3cr1^−/−^* mice had significantly increased thrombus size and delayed thrombus lysis compared to wild-type mice. CX3CL1- and CX3CR1-positive cells were expressed in thrombus tissue. The expression of *Mmp2*, *Plau*, *Plat*, and *Vegf* in thrombus tissue was decreased in *Cx3cr1*^−/−^ mice. This suggested that CX3CL1/CX3CR1 may be a factor required for thrombolysis (our unpublished data).

In summary, CX3CL1/CX3CR1 is involved in the progression of atherosclerosis and may be an effective therapeutic target, but no effective therapeutic agents have been developed. In addition, considering the mechanism of atherosclerosis formation, inhibition of multiple chemokines may be an effective treatment. From a safety perspective, however, chemokine inhibition, like other immunomodulatory therapies, has several drawbacks that should be considered. The resulting chronic monocyte suppression may have significant undesirable effects. Inhibition of CX3CL1/CX3CR1 in a mouse model of deep vein thrombosis delays thrombolysis and may have the side effect of exacerbating DVT.

### 3.2. Coronary Artery Disease

The progression of coronary atherosclerosis leads to plaque rupture and the development of acute coronary syndromes such as stable angina pectoris or ST-elevation myocardial infarction (STEMI). There are several studies on the association of plaque rupture with CX3CL1- and CX3CR1-expressing cells. Ikejima et al. performed optical coherence tomography in patients with unstable angina pectoris and measured plasma levels of soluble CX3CL1 and CX3CR1 with or without plaque rupture. Soluble CX3CL1 was significantly elevated in unstable angina pectoris patients with ruptured plaques. Multiple logistic regression analysis also showed that in addition to plasma levels of soluble CX3CL1, CD14^+^CD16^+^CX3CR1^+^ monocytes, and CD3^+^CX3CR1^+^ lymphocytes were shown to be independent predictors of plaque rupture [87]. On the other hand, concentrations and mRNA expression levels of CCL2, CCL5, and CX3CL1 have also been evaluated in patients with acute myocardial infarction and unstable angina, patients with stable angina, and patients without coronary heart disease [88]. The concentrations of these chemokines were significantly elevated in acute myocardial infarction and unstable angina pectoris patients than in patients with unstable angina pectoris and without coronary heart disease. These findings suggest that CX3CL1 may be involved in plaque rupture.

Bing et al. showed that in STEMI patients, after primary percutaneous coronary intervention (PCI), CX3CL1 concentration on the day after PCI was inversely correlated with ventricle ejection fraction measurements one month later. They also found that patients with CX3CL1 concentrations above the median on the day after PCI had a higher incidence of major adverse cardiac events than those below the median [89]. Furthermore, when the therapeutic role of anti-CX3CL1 antibody administration was examined by inducing myocardial infarction in mice, survival and cardiac function were significantly improved in the group treated with anti-CX3CL1 antibody after myocardial infarction In vitro experiments also showed that CX3CL1 administration to cardiac fibroblasts enhanced fibroblast proliferation; CX3CL1 treatment in myocardial fibroblasts also enhanced fibroblast proliferation in in vitro experiments [90]. These observations indicate that CX3CL1 contributes to the decline in cardiac function after myocardial infarction and may be a useful therapeutic target after the onset of myocardial infarction.

It has been reported that myocardial biopsies from patients with ischemic cardiomyopathy clearly show elevated expression of ADAM10, with cardiomyocytes and endothelial cells predominantly expressing ADAM10 [91]. Deterioration of cardiac function was significantly prevented, and scar size was reduced when ADAM10 inhibitors were administered to mice with myocardial infarction. ADAM10 is involved in the cleavage of the extracellular segment of CX3CL1 and increases soluble CX3CL1. Therefore, ADAM10 inhibitor treatment markedly decreased plasma levels of CX3CL1 [91]. In other experiments, drug-eluting stents coated with the CX3CR1 antagonist AZ12201182 were used in a porcine model for real-world therapeutic applications [92]. The CX3CR1-coated stent significantly reduced in-stent stenosis by approximately 60% compared to bare metal or polymer-only coated stents without affecting reendothelialization of peri-stent inflammation and monocyte/macrophage accumulation. A phase IIa, randomized, two-arm parallel group, placebo-controlled, double-blind, multi-center trial (The FRACTAL Trial) is currently underway to evaluate the safety and myocardial protection of CX3CL1 inhibitors (KAND567) in myocardial infarction, and it is possible that CX3CL1/CX3CR1 targeted therapies will become available in clinical practice [93] (Table 2).

## 4. Cancers

### 4.1. Prostate Cancer

In recent years, there have been many reports showing that combination therapy is more effective than monotherapy for malignant tumors [94,95,96]. In prostate cancer, the combination of a CX3CR1 inhibitor and capivasertib, an AKT inhibitor, was shown to inhibit prostate cancer progression. Inactivation of phosphatase and tensin homolog deleted from chromosome 10 (PTEN), which is common in prostate cancer, increases acetylation of KLF5 through phosphorylated AKT (p-AKT) activation, which stimulates inflammatory cancer-associated fibroblasts (iCAFs) through TNF-α to release FGF9, which in turn activated FGFR1 signaling in prostate cancer cells. CX3CR1 was required for FGF9 to activate FGF receptor 1 (FGFR1) signaling. Inhibition of CX3CR1 sensitized PTEN-deficient prostate cancer to the AKT inhibitor, capivasertib [97].

CX3CL1/CX3CR1 is an important factor in bone metastasis of prostate cancer because prostate cancer cells express CX3CR1 and bone marrow endothelial cells and osteoblasts express CX3CL1 [98]. Liu et al. showed that serum CX3CL1 was significantly higher in serum samples from patients with spinal cord metastasis compared to normal controls in lung, renal, and prostate cancers [99]. They also reported that prostate cancer tissue with spinal cord metastasis overexpressed CX3CR1 more than localized prostate cancer. Overexpression of CX3CR1 induced cell proliferation, migration and invasion, and inhibited cell apoptosis.

Furthermore, the Src/FAK pathway was activated by CX3CL1, and its phosphorylation was dependent on Tyr992 residue of epidermal growth factor receptor (EGFR). Inhibitors of these kinases suppressed cell migration induced by overexpression of CX3CL1 or CX3CR1.

Furthermore, overexpression of CX3CR1 induced spinal cord metastasis of prostate cancer in an in vivo mouse model [100]. Several other in vitro studies have reported the function of CX3CL1/CX3CR1 under hypoxic conditions. CX3CL1/CX3CR1 induced EMT and migration and invasion of androgen-independent prostate cancer cells via TACE/TGF-α in several prostate cancer cell lines under hypoxic conditions [101]. Furthermore, CX3CR1 expression was significantly increased under hypoxic conditions and was involved in the migration and invasion of prostate cancer cells through the action of HIF-1 and NF-κB [102]. Hypoxia causes upregulation of CX3CL1 secretion and expression and enhances cell proliferation by promoting cell cycle progression in prostate cancer cells [103]. Thus, CX3CL1/CX3CR1 is involved in prostate cancer metastasis and may be a target for prostate cancer therapy.

However, the recurrence rate of prostate cancer in patients after prostatectomy was lower with higher expression of CX3CL1 in prostate tissue. Therefore, further studies are needed in the future [104].

### 4.2. Renal Cell Carcinoma

In renal carcinoma, von Hippel–Lindau (VHL) gene deficiency is an important factor in the development of ccRCC and promotes HIF stabilization. CX3CL1 is highly expressed in human ccRCC tumors and is associated with Vhl deficiency. Deletion of CX3CL1 in cancer cells reduces myeloid cell infiltration associated with Vhl deficiency, and Vhl deficiency may contribute to altered immune status [105]. In a study using tissue from a patient with renal cell carcinoma, CX3CL1 in the red bone marrow of the spinal trabecular bone was shown to enhance the migration and invasive potential of renal cell carcinoma cells and to promote the metastasis of renal cell carcinoma to the spine. The migration and invasion of these renal cell carcinoma cells were dependent on Src/FAK activation [106].

On the other hand, there are reports that CX3CL1 acts as a tumor suppressor and is involved in the development of ccRCC. Patients with ccRCC with high CX3CL1 expression had better clinical outcomes than those with low expression. In epigenetic studies, CX3CL1 expression levels were closely associated with the level of CD8^+^ T cell infiltration into the tumor microenvironment (TME). Overexpression of CX3CL1 inhibited tumor cell growth and metastasis in ccRCC and promoted tumor susceptibility to ferroptosis [107]. In a mouse model of RCC, treatment with cabozantinib, a therapeutic agent for renal cancer, significantly increased neutrophil and T-cell infiltration into the tumor and antitumor function. In addition, the expression of CX3CL1, a T-cell-associated chemokine, was increased along with CCL8 in the tumor microcirculation after cabozantinib treatment. These results suggest that cabozantinib may be a promising therapeutic agent in combination with T-cell therapy and other immunotherapies [108].

### 4.3. Lung Cancer

The relationship between bone metastasis and CX3CL1 in lung cancer patients has been reported in several studies [109,110,111]. Serum CX3CL1 and CX3CR1 levels in the bone metastasis group of primary lung cancer were significantly higher than those in lung cancer patients without bone metastasis or healthy controls [109]. In a study using peripheral blood from patients with non-small cell lung cancer (NSCLC), CX3CL1 was significantly higher in patients with bone metastases, and a prediction model using the serum levels of CX3CL1 and CCL28 showed robust predictive accuracy and validity for bone metastases in NSCLC patients [110]. MiR-497-5p, a tumor suppressor, also suppressed cancer cell growth and invasion in NSCLC. MiR-497-5p bound directly to the 3′-UTR of CX3CL1 mRNA, post-transcriptionally repressed its expression, and inactivated its downstream oncogenic pathway, ERK/AKT [112].

Although CX3CL1 has been implicated in bone metastasis and cancer cell growth and invasion, a different result has been reported in relation to overall survival. In a report analyzing lung cancer data from the Gene Expression Omnibus database and The Cancer Genome Atlas, CX3CL1 mRNA expression in lung adenocarcinoma tissue was decreased compared to healthy controls [113]. Furthermore, a pooled analysis based on survival analysis of lung adenocarcinoma patients showed that high CX3CL1 levels were a predictor of good prognosis, but not a significant prognostic factor in squamous cell lung cancer. Genes whose expression levels correlated with CX3CL1 expression were subjected to enrichment analysis, and the LUAD data identified “cell adhesion molecule (CAM)”, “leukocyte transendothelial migration”, and “natural killer cell-mediated cytotoxicity” as the most important biological processes [113]. In another report, high CX3CL1 level was a poor prognostic factor in lung adenocarcinoma patients with a history of smoking, but not in lung adenocarcinoma patients without a history of smoking or in lung squamous cell carcinoma patients with a history of smoking. High expression of CX3CL1 promoted nodal metastasis by activating JNK and MMP2/MMP9 activity in lung cancer cells [114]. These facts make it difficult to accurately predict prognosis based on CX3CL1 alone, and it is important to take into account differences in smoking history and patient background.

### 4.4. Colorectal Cancer

In a report using mesenchymal stem cells (MSCs) for the treatment of colorectal cancer, MSCs induced CX3CR1-high expressing macrophages and promoted M1 polarization, which inhibited tumor growth by promoting CX3CL1 secretion [115]. Furthermore, MSCs improved sensitivity to anti-PD-1 therapy in colorectal cancer by inhibiting PD-1 expression on CD8^+^ T cells and promoting proliferation of CD8^+^ T cells by promoting M1 macrophage polarization. Therefore, the combination of MSCs and anti-PD-1 antibody may be a therapeutic approach for colorectal cancer via the CX3CR1/CX3CL1 axis [115]. Differential gene profiling analysis of tumor vs. non-tumor samples from surgical colorectal cancer patients showed that CX3CL1 and CX3CR1 were significantly upregulated in tumors. Co-expression of CX3CL1 and CX3CR1 by tumor cells was significantly associated with longer disease-free and disease-specific survival. Conversely, axis-negative tumors (low expression of both ligand and receptor) had an increased risk of tumor recurrence and an increased likelihood of metachronous metastasis. Lack of or low levels of CX3CL1-CX3CR1 expression by tumor cells identifies colorectal cancer patients at high risk for metastatic progression [116].

Measuring chemokines and cytokines in plasma of patients with colorectal cancer, high CX3CL1 levels were one of the factors significantly associated with increased overall and cancer-specific mortality [117]. CX3CL1 is expected to be an independent biomarker predicting mortality risk in other carcinomas as well [113,114,117] (Table 3).

### 4.5. Hematological Cancer

The CX3CL1/CX3CR1 axis has emerged as a key player in the tumor microenvironment of various hematologic malignancies. In multiple myeloma (MM), CX3CR1 contributes to tumor progression, osteoclast activation, and angiogenesis [118,119,120]. Meanwhile, in pediatric acute myeloid leukemia (AML), it has been identified as an independent adverse prognostic marker, especially in the context of hyperleukocytosis [121].

CX3CR1 is expressed in some multiple myeloma (MM) cell lines, where stimulation with CX3CL1 has been shown to activate Akt and ERK1/2 signaling pathways and enhance cell adhesion. Furthermore, conditioned media from CX3CL1-treated RPMI-8226 cells significantly promoted the differentiation of RAW264.7 osteoclast precursor cells, and this effect was suppressed by a neutralizing anti-CX3CL1 antibody. These findings suggest that the CX3CL1/CX3CR1 axis contributes to MM progression and may represent a potential therapeutic target [118].

Another study focused on the distribution of monocyte subsets in the bone marrow of MM patients and found that the proportion of non-classical CD16^+^CD14^dim^ monocytes increases in parallel with tumor burden [119]. These monocytes highly express the proinflammatory chemokine receptor CX3CR1 and, upon stimulation with nucleic acids derived from apoptotic tumor cells (via TLR8 agonists), produce cytokines such as TNF-α, IL-6, and CCL3, which are known to promote MM cell proliferation. These results suggest that non-classical monocytes act as tumor-supportive immune cells within the myeloma microenvironment and may also serve as potential therapeutic target [119].

Marchica et al. [120] identified the CX3CL1/CX3CR1 axis as a novel pro-angiogenic factor in the bone marrow of MM patients. CX3CL1 is primarily produced by endothelial cells in a TNF-α–dependent manner, and it promotes angiogenesis through interactions with CX3CR1-expressing monocytes and endothelial cells. These findings highlight the therapeutic relevance of targeting this axis in MM [120].

In pediatric acute myeloid leukemia (AML), hyperleukocytosis is associated with poor prognosis and remains a major therapeutic challenge. Mei et al. conducted a comprehensive analysis using the TARGET database and identified CX3CR1 as an independent prognostic factor among differentially expressed genes [121]. High CX3CR1 expression was associated with reduced overall survival and demonstrated strong physical interaction with CX3CL1, as well as involvement in immune response pathways. Furthermore, CX3CR1 was highly expressed in monocytes, resting NK cells, and CD8^+^ T cells, and was correlated with immune cell infiltration and immune signatures. This study highlights the potential of CX3CR1 as a novel prognostic biomarker and immunotherapeutic target in pediatric AML with hyperleukocytosis, providing valuable insight for the advancement of personalized treatment strategies [121].

## 5. Future Research

The CX3CL1/CX3CR1 axis is intricately involved in a variety of diseases, including renal disease, coronary artery disease, and malignancies, through the regulation of inflammatory leukocyte recruitment (such as macrophages, CD8^+^ T cells, and NK cells), cell survival, cytotoxic immune responses, and tissue fibrosis (Figure 2). Emerging evidence indicates that the effects of this axis can be either protective or detrimental, depending on the disease context, underscoring the need to elucidate disease-specific regulatory mechanisms.

In the field of cancer immunotherapy, CAR-T cells engineered to co-express CX3CR1 (NKG2D-CX3CR1 CAR-T) have been shown to enhance tumor infiltration and antitumor efficacy, demonstrating robust therapeutic effects, especially in tumor models with high CX3CL1 expression [122]. Additionally, the therapeutic potential of targeting this axis is being substantiated in clinical settings, such as the FRACTAL trial, which is evaluating the CX3CR1 antagonist KAND567 in patients with myocardial infarction [93].

However, there remain numerous unresolved challenges in the therapeutic application of CX3CR1. These include a lack of understanding regarding how CX3CR1 expression is regulated within the tumor microenvironment, how to sustain the functional activity of CX3CR1^+^ tumor-infiltrating lymphocytes, and how these cells interact with immunosuppressive components such as myeloid-derived suppressor cells. For CAR-T cell therapy, it is essential to assess long-term safety and clinical efficacy in humans. Moreover, identifying the cancer types and patient subgroups that would benefit most from CX3CR1-engineered CAR-T cells, developing reliable biomarkers, and optimizing combination strategies with IL-15 or immune checkpoint inhibitors remain key areas for future investigation. In renal inflammatory diseases and cardiovascular disorders, the involvement of the CX3CL1/CX3CR1 axis is gradually being elucidated; however, many aspects remain unclear, including its organ-specific functions, its relationship with chronic inflammation, and the long-term effects of therapeutic intervention.

In conclusion, the CX3CL1/CX3CR1 axis represents a common pathological mechanism across multiple disease areas. Its deeper understanding and therapeutic exploitation hold significant promise for the development of novel treatment strategies in cancer, cardiovascular, and inflammatory diseases. Continued basic and clinical research will be crucial in defining both the utility and limitations of targeting this axis and may ultimately contribute to overcoming some of the most challenging diseases in medicine.

## Figures and Tables

**Figure 1 ijms-26-05352-f001:**
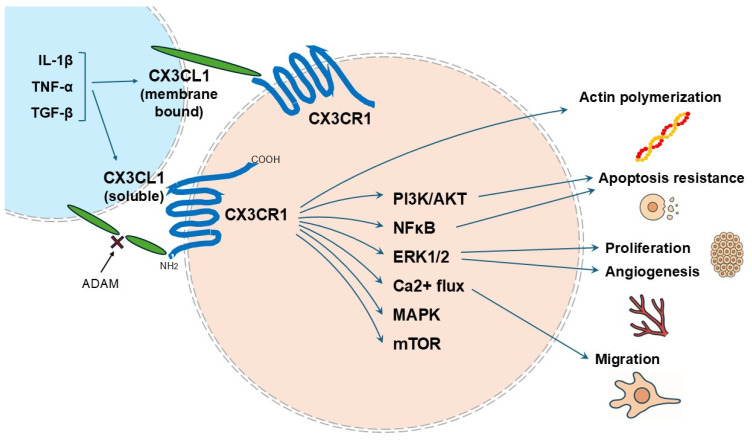
CX3CL1 and CX3CR1 structure, function, and signaling pathways in various pathologies, including kidney disease, cardiovascular disease, and cancer. Upon tissue injury, CX3CL1 is activated by upstream signals such as IL-1β, TNF-α, and TGF-β. CX3CL1 is cleaved by ADAM to form soluble CX3CL1, which activates CX3CR1 as a chemotactic factor for inflammatory cells. Membrane-bound CX3CL1 also promotes adhesion to leukocytes and is involved in inflammatory responses. CX3CL1 contributes to pathways that induce various downstream signaling cascades, including PI3K/AKT, NF-κB, ERK1/2, MAPK, and mTOR. These pathways affect actin polymerization, apoptosis resistance, cell proliferation, angiogenesis, and migration.

**Figure 2 ijms-26-05352-f002:**
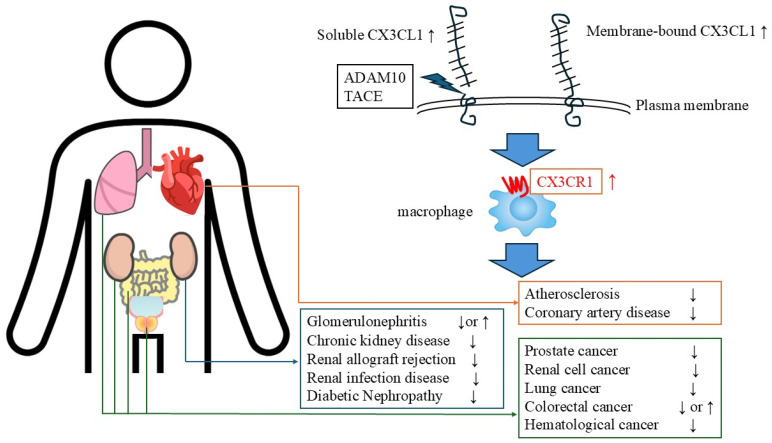
Relationship between diseases and the CX3CL1-CX3CR1 axis (↓: poor prognosis. ↑: better prognosis).

**Table 1 ijms-26-05352-t001:** Relationship between renal disease and the CX3CL1-CX3CR1 axis.

Disease Type	Function	Prognosis with Increased Expression of CX3CL1 CX3CR1 Axis	Reference
Glomerulonephritis	In patients with IgA nephropathy, plasma levels of fractalkine were found to correlate with serum creatinine, 24 h urinary protein excretion, mesangial hypercellularity, infiltration of CD68-positive macrophages and CD20-positive B cells in renal tissue, as well as overall renal prognosis.	↓	[30]
	Increased expression of fractalkine in the glomeruli and urine, associated with activation of the CX3CR1–fractalkine signaling pathway, may contribute to the worsening of gross hematuria.	↓	[31]
	Increased levels of urinary fractalkine tended to be associated with good prognosis with crescentic glomerulonephritis	↑	[32]
	MPO-ANCA increases CX3CL1 expression on human glomerular endothelial cells (HGECs) and promotes the recruitment of CD16^+^ monocytes to the kidney.	↓	[33]
	CX3CL1 mRNA was increased in the glomerular and tubular interstitium of patients with renal inflammation.	↓	[34]
	CX3CL1 expression and CD16^+^ monocyte accumulation in glomeruli were associated with histopathological features of glomerular lesions in MRL/lpr mice	↓	[38]
	Glomerulonephritis in MRL/lpr mice is influenced by CX3CR1 through a mechanism dependent on gut microbial composition.	↑	[39]
	A significantly disrupted gut microbiota, along with a compromised intestinal barrier, was observed in the context of Cx3cr1 deficiency.	↑	[40]
Chronic kidney disease	CKD patients exhibited higher plasma CX3CL1 levels, which were found to be negatively associated with eGFR.	↓	[41,42]
	Fibrotic kidneys exhibited CX3CR1 presence in various cell populations, such as mononuclear cells, epithelial tubule cells, dendritic cells, and α-SMA/vimentin-positive interstitial myofibroblasts.	↓	[46]
	CD1c+ DCs as the predominant source of profibrotic TGF-β and highest expressors of the fractalkine receptor CX3CR1 within the renal DC compartment.	↓	[48]
	CX3CR1^−/−^ mice showed significantly reduced collagen deposition and macrophage infiltration in kidney.	↓	[50]
	CX3CL1/CX3CR1 axis controls the survival of Ly6C^−^CX3CR1^high^ macrophages	↓	[52]
	In the absence of CX3CR1, renal macrophage numbers were increased, along with enhanced production of TGF-β, a major profibrotic cytokine. This accumulation resulted from elevated local proliferation, even though monocyte recruitment was diminished and apoptotic activity was higher in renal tissue.	↑	[53]
Renal allograft rejection	Pati Elevated serum CX3CL1 concentrations before transplantation were observed in patients with acute kidney allograft rejection, implying an increased proinflammatory status relative to individuals with favorable outcomes.	↓	[59]
	The levels of fractalkine on day 0 of acute renal rejection group was significantly higher than that in no renal rejection group.	↓	[55]
	Acute renal rejection patients had significantly higher serum fractalkine levels compared to levels observed in the no renal rejection group and healthy controls.	↓	[56]
	CX3CL1 may have a functional role in leucocyte adhesion and retention, at selected tubular sites in acute renal inflammation.	↓	[57]
	Cells positive for CX3CR1 largely comprised CD68-expressing monocytes/macrophages and dendritic cells positive for CD209/DC-SIGN. The percentage area of CX3CR1 positivity was associated with steroid responsiveness and was predictive of worse clinical prognosis at 3 and 12 months post-transplantation.	↓	[58]
Renal infection diseases	CX3CR1-dependent infiltration of Ly6C(high) inflammatory monocytes into the kidney was accompanied by altered cell motility and increased adhesion to the renal vascular wall.	↑	[61]
	Renal macrophage deficiency in infected *Cx3cr1^−/−^* mice was due to reduced macrophage survival, not impaired proliferation, trafficking, or differentiation.	↑	[65]
Diabetic Nephropathy	In patients with diabetic kidney disease, Gene Expression Omnibus data analysis revealed that CX3CR1 expression inversely correlated with estimated glomerular filtration rate.	↓	[71]
	Higher CX3CL1 level also was associated with prevalent diabetes in adjusted models.	↓	[73]
	CX3CL1-CX3CR1 is a novel inflammatory adipose chemokine system that modulates monocyte adhesion to adipocytes and is associated with obesity, insulin resistance, and type 2 diabetes.	↓	[74]
	In the early phases of diabetic kidney disease, increased expression of CX3CL1 and CX3CR1 was observed. CX3CL1 was prominently stained in diabetic kidneys, notably within the glomerular capillary lumens and peritubular capillaries. Only a small number of CX3CR1-positive cells were detected infiltrating the diabetic glomeruli.	↓	[75]

↓: poor prognosis. ↑: better prognosis.

**Table 2 ijms-26-05352-t002:** Relationship between cardiovascular disease and the CX3CL1-CX3CR1 axis.

Disease Type	Function	Prognosis with Increased Expression of CX3CL1 CX3CR1 Axis	Reference
Atherosclerosis	The number of CX3CL1-expressing cells positively correlates with the number of CX3CR1-positive cells in human carotid artery plaques.	↓	[77]
	Soluble CX3CL1 levels in circulating blood were significantly elevated in the presence of severe stenosis of the carotid artery	↓	[78]
	CX3CL1^−/−^ mice had significantly reduced macrophage accumulation and atherosclerotic lesion size compared to mice lacking each gene alone.	↓	[79]
	The joint inhibition of CCL2, CX3CR1, and CCR5 in ApoE-deficient mice led to the suppression of monocytosis in the bone marrow and lowered circulating monocyte levels, despite persistent hypercholesterolemia.	↓	[80]
	The 249I allele carriers of the CX3CR1 V249I polymorphism had a reduced risk of atherosclerosis and coronary artery disease in the heterozygous state.	↓	[81]
Coronary artery disease	The plasma levels of soluble CX3CL1 were significantly increased in UAP patients with plaque rupture.	↓	[86]
	The concentration of CX3CL1 was significantly elevated in acute myocardial infarction and unstable angina pectoris patients than in patients with unstable angina pectoris and without coronary heart disease	↓	[87]
	STEMI patients after primary percutaneous coronary intervention, CX3CL1 concentration on the day after PCI was inversely correlated with ventricle ejection fraction measurements one month later.	↓	[88]
	Survival and cardiac function were significantly improved in the group treated with anti-CX3CL1 antibody after myocardial infarction.	↓	[89]
	When ADAM10-mediated cleavage of CX3CL1 is abolished, IL-1β-driven inflammation is suppressed, neutrophil release from the bone marrow is decreased, and infiltration into myocardial tissue is limited.	↓	[90]
	CX3CR1 represents a promising therapeutic target to inhibit monocyte adhesion and inflammation as well as in-stent neointimal hyperplasia, while preserving stent re-endothelialization.	↓	[91]

↓: poor prognosis. ↑: better prognosis.

**Table 3 ijms-26-05352-t003:** Relationship between cancer and the CX3CL1-CX3CR1 axis.

Cancer Type	Function	Prognosis with Increased Expression of CX3CL1 CX3CR1 Axis	Reference
Prostate cancer	CX3CR1 was required for FGF9 to activate FGF receptor 1 (FGFR1) signaling.	↓	[96]
	Targeting CX3CL1 and its influence on the EGFR, Src, and FAK signaling pathways could provide new avenues for early intervention against spinal metastases in prostate cancer.	↓	[99]
	CX3CL1/CX3CR1 induces EMT and migration and invasion of androgen-independent prostate cancer cells through TACE/TGF-α/EGFR pathway activation.	↓	[100]
	Hypoxia-induced CX3CR1 expression requires both HIF-1 and NF-κB and is linked to increased movement and invasiveness of prostate cancer cells.	↓	[101]
	The activation of CX3CR1 in prostate cancer cells recruits an important antiapoptotic signaling pathways such as Akt/GSK3.	↓	[97]
	Under hypoxic conditions, CX3CL1 secretion and expression were increased, resulting in enhanced proliferation of prostate cancer cells through stimulation of the cell cycle.	↓	[102]
Renal cancer	The chemokine CX3CL1 was highly expressed in human ccRCC tumors and was associated with Vhl deficiency	↓	[104]
	Elevated levels of CX3CL1 suppressed both proliferation and metastatic potential of tumor cells while enhancing their sensitivity to ferroptosis in ccRCC	↑	[106]
	CX3CL1 in the red bone marrow of spinal cancellous bone enhances migration and invasion abilities of RCC cells	↓	[105]
	Cabozantinib treatment induced expression of CX3CL1 as T cell-related chemokines in the tumor microenvironment	↑	[107]
Lung cancer	CX3CL1 was elevated in patients with bone metastases in NSCLC patients.	↓	[108,109]
	MiR-497-5p expression is reduced in both NSCLC tissues and cell lines, where it suppresses tumor proliferation and invasion by downregulating CX3CL1 and subsequently inhibiting the ERK/AKT signaling pathway.	↓	[111]
	Elevated CX3CL1 mRNA expression in lung adenocarcinoma tissues was associated with better patient outcomes, potentially through mechanisms involving cell adhesion molecules, leukocyte transendothelial migration, and NK cell cytotoxic activity.	↑	[112]
Colorectal cancer	Mesenchymal stem cells recruits CX3CR1^high^ macrophages and promotes M1 polarization to inhibit tumor growth via highly secretion of CX3CL1.	↑	[114]
	The CX3CL1-CX3CR1 chemokine axis expressed by tumors acts to retain cells by enhancing adhesion between similar cells, thereby restricting tumor dissemination to distant metastatic locations.	↑	[115]
	High plasma CX3CL1 level was an independent poor prognostic factor.	↓	[116]
Hematological cancer	CX3CR1 is expressed in certain multiple myeloma cell lines, where stimulation with CX3CL1 activates the Akt and ERK1/2 signaling pathways and enhances cell adhesion. The differentiation of osteoclast precursor cells was inhibited by a neutralizing anti-CX3CL1 antibody.	↓	[117]
	In multiple myeloma, the proportion of CX3CR1-high non-classical monocytes (CD16^+^CD14^dim^) increases with tumor burden, and these cells produce tumor-promoting cytokines such as TNF-α, IL-6, and CCL3 in response to stimuli derived from apoptotic tumor cells.	↓	[118]
	The CX3CL1/CX3CR1 axis promotes bone marrow angiogenesis in multiple myeloma by mobilizing CX3CR1-positive monocytes and endothelial cells in response to TNF-α–dependent CX3CL1 production.	↓	[119]
	In pediatric AML, hyperleukocytosis is associated with poor prognosis, and high CX3CR1 expression is linked to reduced survival through its interaction with CX3CL1 and involvement in immune response pathways, as well as strong associations with immune cell infiltration.	↓	[120]

↓: poor prognosis. ↑: better prognosis.

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
