# Peer review of "Pathophysiological Roles of the CX3CL1-CX3CR1 Axis in Renal Disease, Cardiovascular Disease, and Cancer"

_ijms, 2025, doi:10.3390/ijms26115352_

Round 1

Reviewer 1 Report

Comments and Suggestions for Authors

In the manuscript titled "Pathophysiological Roles of the CX3CL1-CX3CR1 Axis in Renal Disease, Cardiovascular Disease, and Cancer," a research team led by Iwahashi and Kondo discusses the findings on the pathophysiological role of the CX3CL1-CX3CR1 axis in these diseases. 

Overall, the authors provide a good summary and present their findings in a well-structured manner. However, there are a few minor weaknesses that need to be addressed. 

1.References are needed in the following locations: Lines 67-69, 73-75, 77-85, 159-162, 182-184, 247-248, 507-511, and 525-527. 

2.The authors may want to consider discussing the functions and mechanisms of CX3CL1 in hematological malignancies, such as multiple myeloma and leukemia. 

3.It would also be beneficial to include more information on the clinical value of CX3CR1 as a therapeutic target, including any treatment strategies or clinical trials related to the CX3CL1-CX3CR1 axis, as well as any correlations with CAR T-cell. Reference PMID37070068 may be helpful in this regard.

Author Response

Reviewer 1

  1. References are needed in the following locations: Lines 67-69, 73-75, 77-85, 159-162, 182-184, 247-248, 507-511, and 525-527. 

We thank the reviewer for the valuable comment. In response to the comments, we have carefully revised the manuscript and added appropriate references to support the statements made in the specified sections. We hope that the reviewers would find these additions suitable for the publication.

Line (67-69) 85-88, Ref. [26]

Line (73-75) 93-94, Ref. [27]

Line (77-85) 97-106, Ref. [28]

Line (159-162) 182-184, Ref. [40]

Line (182-184) 203-204, Ref. [47]

Line (247-248) 268-269, Ref. [62]

Line (507-511) 528-532, Ref. [113]

Line (525-527) 546-548, Ref. [115]

  1. The authors may want to consider discussing the functions and mechanisms of CX3CL1 in hematological malignancies, such as multiple myeloma and leukemia. 

In accordance with the comments, we have added a new section describing the roles of CX3CL1 in hematological malignancies such as multiple myeloma and leukemia together with some references (line 566-602).

3.
 It would also be beneficial to include more information on the clinical value of CX3CR1 as a therapeutic target, including any treatment strategies or clinical trials related to the CX3CL1-CX3CR1 axis, as well as any correlations with CAR T-cell. Reference PMID37070068 may be helpful in this regard.

In response to the comments, we have added a new paragraph describing the clinical relevance and future perspectives of CX3CR1 as a therapeutic target (line 607-620). In particular, we have discussed preclinical studies, including the enhanced antitumor efficacy of CAR-T cells co-expressing CX3CR1 in solid tumor models with high CX3CL1 expression. Furthermore, we have incorporated information on ongoing clinical trials evaluating CX3CL1/CX3CR1-targeted agents, such as the CX3CR1 antagonist KAND567 (FRACTAL trials). We also highlight the remaining challenges and potential future directions in both malignant and non-malignant diseases (line 621-633).

Reviewer 2 Report

Comments and Suggestions for Authors

Reviewer Comments on the Manuscript: "Pathophysiological Roles of the CX3CL1-CX3CR1 Axis in Renal Disease, Cardiovascular Disease, and Cancer"

The manuscript provides a comprehensive and well-structured review of the current understanding of the CX3CL1-CX3CR1 axis and its involvement in the pathogenesis and progression of inflammatory diseases and malignancies. The authors successfully summarize the recent advances in the field, with a particular emphasis on renal disease, cardiovascular disease, and cancer. The review is informative and highlights the multifaceted roles of this chemokine-receptor pair in mediating leukocyte migration and inflammatory responses.

Minor Comments:

  1. The introduction would benefit from the inclusion of recently published studies (within the last 2–3 years) to better contextualize the relevance of CX3CL1-CX3CR1 signaling in contemporary research.
  2. The addition of a schematic figure summarizing the key signaling pathways and mechanisms linking the CX3CL1-CX3CR1 axis to renal, cardiovascular, and cancer pathologies would significantly enhance the clarity and visual appeal of the manuscript.
  3. The manuscript would be strengthened by a dedicated section highlighting key open questions and potential future research directions in the study of the CX3CL1-CX3CR1 axis. This could guide researchers toward unresolved issues and emerging avenues in the field.

Author Response

Reviewer 2

Minor Comments:

1.The introduction would benefit from the inclusion of recently published studies (within the last 2–3 years) to better contextualize the relevance of CX3CL1-CX3CR1 signaling in contemporary research.

2.The addition of a schematic figure summarizing the key signaling pathways and mechanisms linking the CX3CL1-CX3CR1 axis to renal, cardiovascular, and cancer pathologies would significantly enhance the clarity and visual appeal of the manuscript.

In respond to the Comments 1 & 2, we have described recent topics with a schematic diagram (line 62-81 and New Fig. 1). I believe that this allows for a clearer understanding of the major signaling pathways linking CX3CL1-CX3CR1 interaction to renal, cardiovascular, and cancer pathologies and their importance in modern research.

3.The manuscript would be strengthened by a dedicated section highlighting key open questions and potential future research directions in the study of the CX3CL1-CX3CR1 axis. This could guide researchers toward unresolved issues and emerging avenues in the field.

In accordance with the comment, we have added a new paragraph summarizing key unresolved issues and future research directions related to the CX3CL1-CX3CR1 axis “5. Future research” (line 607-640).